# Bioactivity Performance of Pure Mg after Plasma Electrolytic Oxidation in Silicate-Based Solutions

**DOI:** 10.3390/molecules26072094

**Published:** 2021-04-06

**Authors:** Yevheniia Husak, Joanna Michalska, Oleksandr Oleshko, Viktoriia Korniienko, Karlis Grundsteins, Bohdan Dryhval, Sahin Altundal, Oleg Mishchenko, Roman Viter, Maksym Pogorielov, Wojciech Simka

**Affiliations:** 1Medical Institute, Sumy State University, 40018 Sumy, Ukraine; evgenia.husak@gmail.com (Y.H.); oleshkosanya007@gmail.com (O.O.); vicorn77g@gmail.com (V.K.); drigval007@gmail.com (B.D.); 2Faculty of Chemistry, Silesian University of Technology, 44-100 Gliwice, Poland; joanna.k.michalska@polsl.pl; 3Institute of Atomic Physics and Spectroscopy, University of Latvia, LV-1586 Riga, Latvia; karlis.grundsteins@gmail.com (K.G.); sahin.altundal@edu.rtu.lv (S.A.); 4Faculty of Materials Science and Applied Chemistry, Riga Technical University, LV-1048 Riga, Latvia; 5NanoPrime, 39-200 Dębica, Poland; dr.mischenko@icloud.com; 6Zaporizhzhia State Medical University, 26 Prosp. Mayakovskogo, 69035 Zaporizhzhia, Ukraine

**Keywords:** magnesium, plasma electrolytic oxidation, silicate bath, degradation rate, biocompatibility, antibacterial properties

## Abstract

The biodegradable metals, including magnesium (Mg), are a convenient alternative to permanent metals but fast uncontrolled corrosion limited wide clinical application. Formation of a barrier coating on Mg alloys could be a successful strategy for the production of a stable external layer that prevents fast corrosion. Our research was aimed to develop an Mg stable oxide coating using plasma electrolytic oxidation (PEO) in silicate-based solutions. 99.9% pure Mg alloy was anodized in electrolytes contained mixtures of sodium silicate and sodium fluoride, calcium hydroxide and sodium hydroxide. Scanning electron microscopy (SEM), energy-dispersive X-ray spectroscopy (EDX), contact angle (CA), Photoluminescence analysis and immersion tests were performed to assess structural and long-term corrosion properties of the new coating. Biocompatibility and antibacterial potential of the new coating were evaluated using U2OS cell culture and the gram-positive Staphylococcus aureus (*S. aureus*, strain B 918). PEO provided the formation of a porous oxide layer with relatively high roughness. It was shown that Ca(OH)_2_ was a crucial compound for oxidation and surface modification of Mg implants, treated with the PEO method. The addition of Ca^2+^ ions resulted in more intense oxidation of the Mg surface and growth of the oxide layer with a higher active surface area. Cell culture experiments demonstrated appropriate cell adhesion to all investigated coatings with a significantly better proliferation rate for the samples treated in Ca(OH)_2_-containing electrolyte. In contrast, NaOH-based electrolyte provided more relevant antibacterial effects but did not support cell proliferation. In conclusion, it should be noted that PEO of Mg alloy in silicate baths containing Ca(OH)_2_ provided the formation of stable biocompatible oxide coatings that could be used in the development of commercial degradable implants.

## 1. Introduction

Titanium and its alloys are the most popular solution for orthopedics applications due to high biocompatibility, low corrosion and relatively high clinical success [1]. A mismatch between the mechanical properties of the implant and those of the surrounding tissue, leading to stress-shielding [2] and eventually to bone resorption and possible release of toxic elements during long-term exploitation [3] are the main disadvantages of Ti application. The biodegradable metals (Zn, Mg and Fe) are a convenient alternative to conventionally used metals [4]. Due to low toxicity, Mg is a promising candidate for biodegradable implant development [5]. It is the fourth most abundant cation in the human body and can be completely utilized by different pathways after release from the implant [6]. Application of Mg implant makes unnecessary a second surgical procedure and significantly decreases the treatment cost and minimizes complication rate [7]. However, wide application of Mg implant is limited due to its high chemical reactivity and fast corrosion in a physiological environment that leads to the loss of mechanical integrity [8]. The development of new Mg-based alloys and searching for more effective methods of surface layer modification are two main strategies that could overcome the abovementioned complications [6]. As we demonstrated in a recent review, the addition of different alloying elements (Ca, Al, Zn, Zr, Sr, etc.) did not improve corrosion of Mg and led to the release of toxic metals during exploitation [9]. Some recent reviews demonstrated that the formation of a barrier coating on Mg alloys could be a successful strategy for the production of a stable external layer that prevents fast corrosion [10].

Taking into account that Mg is highly reactive in an acidic environment [11], there are limited technologies that could be used to produce a compact coating and do not affect Mg substrate at the same time [12]. In addition, the anti-corrosion function, coating could increase biocompatibility and provide some other beneficial aspects such as antibacterial effect. Two different strategies—organic and inorganic coatings are used to prevent implant corrosion [6]. Chitosan, alginate, cellulose, collagen and gelatin are used to form biocompatible coatings and demonstrate high biocompatibility and improved corrosion resistance [13,14,15]. The main disadvantage of organic coating is pure mechanical stability and possible damage during the implantation stage. Controlled reactions between the substrate and the environment with the simultaneous transformation of the substrate’s surface into an inorganic oxide layer are the more suitable method to provide a stable anticorrosive layer. Anodization or chemical conversion are the conventional examples that can produce a protective coating on a magnesium substrate [16]. Unfortunately, these methods suffer from low reproducibility and the chemical composition of electrolytes is limited. Plasma electrolytic oxidation (PEO) is a simple, low cost and effective method to provide a ceramic coating on different alloys and demonstrates high reproducibility and effectiveness [17]. There is some data about PEO of pure Mg and its alloys for implant purposes that demonstrate high perspective and require further investigation. The electrolyte bath composition and PEO regimens, including voltage and current density, are still under investigation. Echeverry-Rendon M. et al. used a solution of fluoride, hexamethylenetetramine and mannitol to produce a protective layer on Mg coronary stents. They proved a delay in corrosion and support of vascular cell proliferation. Using a standard Kermasorb^®^ electrolyte for PEO of Mg-RE and Mg-Zn-Ca alloys, demonstrated hemocompatibility with a low rate of platelet adhesion [18]. Dong et al. used an electrolyte consisting of NaH_2_P_2_O_7_, K_2_TiF_6_, NaF, C_6_H_12_N_4_ and NaOH. However, PEO treatment produced cracked coatings with increased corrosion [19]. Feng Peng et al. used fluoride solution for PEO of Mg-Al alloy and provided the formation of stable and porous coating with a high cytocompatibility profile [20]. Other researchers used alkali silicate solution to provide double-stage PEO coating of AZ91 Mg Alloy and demonstrated compact inner layer structure, significant improvement in the corrosion resistance and the mechanical properties of the coating [21]. Thus, the silicate solution could be a promising agent for the PEO treatment of Mg alloys due to the formation of a high specific surface area and better biocompatibility [22]. The addition of Ca in PEO coating could significantly affect degradation and corrosion properties, but the most previous studies demonstrate application of high voltage (between 400 and 500 V) and long treatment times (up to 30 min) that could affect Mg substrate [23]. It is required to optimize PEO parameters to reduce treatment time for Ca-based solution. So, the objective of current research is an elaboration of PEO treatment of pure Mg in different silicate-based solutions and the assessment of structural and biological properties of a new coating.

## 2. Results and Discussion

### 2.1. Coatings Characterization

After plasma electrolytic oxidation, the surface of magnesium alloys has a typical porous structure. PEO in the C1 electrolyte generates the sub-micron pores with size up to 0.5 μm. There are some single pores with a diameter ranging from 1.0 to 5.0 μm, but they did not significantly influence porosity (Figure 1). Increasing PEO voltage from 200 to 250 V led to a significant reduction of pore area, from 21 to 9%. PEO treatment in the C2 electrolyte provided the formation of pores with a mean diameter from 1 to 5 μm with some single pores with size up to 25.0 μm (Figure 1). Pore area in C2 coatings ranged from 15% (250 V) to 12% (300 V). The pore size is one of the critical and controversial points for cell adhesion and proliferation. There is some data suggested that pores sizes from 5 to 70 μm are favorable for osteogenic cell adhesion and proliferation [24]. Brennan C. et al. demonstrated the fast protein adhesion to the nanoporous surface with the following cell attachment [25]. Other studies demonstrated antifouling properties of the nanoporous surface that dramatically decrease bacteria adhesion [26]. However, most researchers suggested that chemical components, wettability and pore size can influence cell and bacteria adhesion simultaneously and each new surface requires detailed analysis.

Mg, O, Si, F were the major elements identified on energy-dispersive spectra in both electrolyte type coatings (Figure 2, Table 1). Clear incorporation of Si into all investigated oxide coatings was observed. A higher concentration of Si was found for the C2-type electrolyte. It was also observed that with increasing anodizing voltage more Si was incorporated into the oxide coating. The incorporation of Ca into the C2-type coatings was equally small and the anodizing voltage did not affect the concentration of Ca in oxide the coating. It could be explained by the formation of calcium fluoride during a long PEO treatment time [24]. However, the amount of incorporated F was significantly higher (0.31–0.43 at. ratio) for the C2-type surface while for the samples treated in the C1-type electrolyte the fluorine content was below 0.1 at. ratio. (Table 1).

The formation of a stable oxide layer is the main parameter that will influence degradation resistance, biocompatibility and clinical outcomes. Some studies demonstrated that thick coating prevents early in vivo corrosion and increases Mg implant biocompatibility [27,28]. The C1 electrolyte in both voltage regimens generated a thin oxide layer with a thickness from 2.06 ± 0.7 to 2.09 ± 0.38 μm. Calcium hydroxide-based electrolyte (C2) led to a dramatic increase of oxide layer thickness that reached up to 14.52 ± 1.8 μm with a voltage of 300 V (Figure 3).

Cross-section mapping confirmed the formation of oxide coatings on the Mg substrate with Si and F incorporation into the oxide layer (Figure 4).

The surface texture has a strong influence on the adhesion cells to the coating [25]. Obtained coatings demonstrated common features of the layers—the presence of porosity. On the other hand, there is a direct relation between applied voltage, type of electrolyte and roughness. The thickness of the coatings and roughness parameters Ra and Rz increased with an increased voltage for the C2-type samples. PEO coatings prepared in the Ca(OH)_2_ -containing electrolyte showed a trend of increasing roughness with the increase of applied voltage compared to the C1 electrolyte. The values of roughness parameters are given in Figure 5.

Another important peculiarity of the Ca-containing coatings was the apparent increase of the hydrophilicity (Figure 5). Wettability is an essential characteristic of the surface that promotes cell adherence in biomedical applications. Hydrophilic surface support deposition and adhesion of proteins [24]. The contact angle of the surface C1 200V and C1 250V was 26.22° and 22.7°, respectively, which means the surfaces had a hydrophilic property. The Ca(OH)_2_ addition to electrolyte reduced the contact angle values. The reduction was approximately 30% for the C2 250V coating. Moreover, the surface of C2 300V had the lowest contact angle (0°) and was highly hydrophilic. It appeared that an increase in applied voltage caused a decline in contact angle. According to these data, surface roughness correlated with contact angle values for C2-type coatings. At lower contact angles, higher surface texture values were facilitated (Figure 5).

The obtained PL spectra are plotted in Figure 6. Mg PEO samples showed four characteristic emission peaks, centered at 360–380, 450, 500 and 660 nm.

MgO nanostructures showed emission in the UV-Vis-NIR range [29,30,31,32,33,34]. The observed emission can be explained by structural defects [29]. PL peaks, observed in the range 380–520 nm, are associated with F/F+ centers (alkali ions/atoms) and oxygen vacancies [29,30,31,32,33,34]. The observed PL emission in the range of 600–800 nm is attributed to surface structural defects, such as oxygen vacancies and interstitials [29,30,31,32,33].

The obtained PL results showed that the PEO process in the C1 electrolyte generated fewer structural defects, compared to the samples modified in the C2 electrolyte. An increase in the voltage resulted in a significant increase of the peak at 660 nm for the C1-type coatings and an insignificant increase of the peak intensity at 450–520 nm for the C2-type coatings.

Correlation of PL data with SEM and EDX results had been performed. It was shown that Ca(OH)_2_ was a crucial compound for oxidation and surface modification of Mg implants, treated with the PEO method. The addition of Ca^2+^ ions resulted in more intense oxidation of the Mg surface and growth of the oxide layer with a higher active surface area. Therefore, defect concentration and higher surface area can explain more intense PL emission for the C2-type coatings.

### 2.2. Long-Term Immersion Test in SBF

Long-term immersion test in SBF solution characterizes the stability of the surface and its ability to induce the precipitation of the calcium phosphate deposits. The weight loss results of the bare Mg alloy and C1-C2 PEO-modified samples are shown in Figure 7.

Until the seventh day of the immersion test, the weight loss for all tested samples (control and PEO-treated) was slight, less than 0.5% and the degree of degradation of the samples was similar. In general, Mg can be oxidized in an SBF environment, releasing hydrogen and depositing of Mg(OH)_2_ precipitates on the sample surface (see Equations (1) and (2) [35]). As the dehydration of Mg(OH)_2_ proceeds, according to Equation (3), the volume of the pre-deposited hydroxide layer possesses a tendency to shrink, resulting in the peeling-offs on the Mg surface. The rate of weight loss for pure Mg (control) is low due to MgO layer formation and magnesium dissolution [36]:2Mg + 2H_2_O = 2Mg^2+^ + 2OH^−^ + H_2_(1)
Mg^2+^ + 2OH^−^ = Mg(OH)_2_(2)
Mg(OH)_2_ = MgO + H_2_O(3)

A clear increase in the degradation of the samples was found after 21 days of the immersion test. However, the PEO-treated samples revealed lower weight losses compared to the Mg control sample but underwent a visible destruction process (Figure 8). It is closely related to the degradation within the hidden closed pores, which are gradually exposed to the electrolyte during the immersion time. MgO, the main constituent of the PEO coating is not stable in an aqueous environment [35] and facilitates the reaction shown in Equation (3) proceeding towards the adverse direction. The C2 250V coating demonstrated the best ability to degradation process compared with other PEO coatings. As a result of 42 and 63 days of immersion, the C2 250V showed the lowest values of weight loss (2.85 ± 0.084% and 4.88 ± 0.098%, respectively). It must be noted that during 63 days of immersion, the weight loss of pure Mg was almost 2 times higher. The high corrosion resistance and decreased degradation in a simulated body fluid was also proved in the case of Mg alloys that were anodized under PEO conditions [37].

Since the calcium phosphate from SBF solution can nucleate both on the coating and at the metal surface simultaneously [38], these precipitates are suggested to be the Ca-P compounds. The surface is gradually sealed by the precipitates and accumulation of corrosion products. The EDS analyses performed at different time intervals during the immersion test proved the deposition of Ca^2+^ and HPO_4_^2−^ ions both on the surface of the control sample and the obtained coatings (Figure 9). PEO oxide coatings generated with the C2-type electrolyte exhibited a higher Ca/P ratio in comparison to the surfaces obtained by using C1-type electrolyte. The highest differences were visible within the first week of the immersion test. After 63 days of the immersion test, the Ca/P ratio for all the C1-type samples and one C2-type sample anodized at 250 V was similar. For this time interval, the Ca/P ratio for the C2 300 V sample was the highest and achieved 1.19. The concentration of Si and F gradually decreased with the time of immersion in SBF, which reflected in the slow degradation process of the coating (data not shown).

### 2.3. Cell Viability

U2OS cells did not attach to the control samples within 24 h after seeding. It was reported previously that corrosion products present on the sample surface can significantly change the pH of the medium and affect cell metabolism [39]. As the days went by, the resazurin reduction decreased significantly which demonstrated the toxic effect of the corrosion products on cell viability. All the PEO modified samples demonstrated adequate cell adhesion with a reassuring reduction from 45.6 ± 6.6 to 53.2 ± 8.1% with no significant difference between groups (Figure 10).

There was significant cell proliferation in the C2-type coatings at the third and the fifth days up to 83.8 ± 7.4%. In contrast to the C2-type coatings, the cells on both C1 200V and C1 250V slightly proliferated on the third day and resazurin reduction significantly decreased up to the fifth day. The proliferation rate at this term was still over the values for the control samples but significantly less than in C2-type coatings. SEM analysis demonstrated that C2-type PEO regimens provided a thicker oxide coating that could protect Mg substrate from corrosion. Some previous research proved that complete precise coating can improve biocompatibility and depends on coating thickness and its stability [38,40]. Fast coating corrosion or structural defects lead to failure in cell attachment and leads to significant toxicity.

### 2.4. Antibacterial Properties

The antibacterial activity of the specimens was evaluated by a time-depending test using *S. aureus*. Figure 11 displays the bacteria cell counts adherent onto the specimens’ surfaces after different assay times. At every time point, the growth of bacteria was reduced on the surface of the C1-type coatings. The results showed that up to 24 h, the bacterial populations on the C1-type coatings enlarged steadily but were significantly lower than those of the control samples. Thus, the initial amount of bacteria was about 3.5 log10 CFU/mL for both C1 200V and C2 250V PEO-treated samples. At 4 h, the growth of *S. aureus* reached for these samples 4.69 log10 CFU/mL and 4.00 log10 CFU/mL, respectively. The bacterial growth increased gradually and more rapidly on the C2-type coatings. Hence, both C2 250V and C2 300V specimens demonstrated an increase in a bacterial amount up to 5.70 log10 CFU/mL. After 6 h of incubation, the colonies reached 6.00 log10 CFU/mL for C1 200 V and 6.7 log10 CFU/mL for C2 250V specimens but were still significantly lower than for the control samples. Otherwise, at 24 h, only sample C1 200V revealed the growth reduction compared to all other PEO treated and control specimens (*p* < 0.0005). As for other investigated samples (PEO-treated and control), there was no reduction in the number of bacteria cells at this time point of the experiment.

The ability to prevent secondary infection due to antiadhesion behavior is vital for biomaterial surface characteristics. Based on the antimicrobial activity assay results, it is suggested that C1-type coatings on magnesium alloys may be acceptable for medical application, as their surfaces did not facilitate initial bacterial adhesion. Despite *S. aureus* biofilm formation enlarged with incubation, a significant difference between the C1-type and control samples in CFU/specimen was confirmed. Moreover, the results showed higher inhibition of bacterial adhesion for the C1-type than the C2-type coatings. The possible explanation is that the material surface characteristics may affect bacterial growth and reduction [41]. The value of surface roughness influences the surface topography and bacterial accumulation occurred on it, consequently [42]. The surface roughness was significantly affected by PEO treatment depending on the composition of the electrolyte bath. The bacteria accumulation is elevated with increasing surface roughness. The bacterial cells preferred to attach to the more developed surfaces as irregular structures comparable to the bacteria’s size provided a larger surface and favorable sites for colonization and biofilm formation [43]. Additionally, more hydrophilic C2-type coatings could accelerate bacterial adhesion providing specific binding sites potentially via the accumulation of the proteins [44].

## 3. Materials and Methods

### 3.1. Materials

The pure Mg (99,99%) was obtained from Polmag (Kędzierzyn-Koźle, Poland). Na_2_SiO_3_, NH_4_F, NaOH, Ca(OH)_2_ were bought from Sigma–Aldrich (St. Louis, MO, USA). The Mg samples in a form of cubes with a 10 × 10 mm face size were used in the experiment. All samples were ground using abrasive paper (Hermes BW114) with 400, 1000, 1200 and 1500 granulation, then were ultrasonically cleaned 2-propanol for 5 min. In the last step, just before PEO treatment, samples were rinsed with distilled water.

### 3.2. Plasma Electrolytic Oxidation (PEO)

To obtain a new bioactive surface, the pure magnesium was modified by PEO. The active surface area was 5 cm^−2^ since one face of a cube was insulated. The samples were subjected to anodization in two different electrolytes with the chemical composition given in Table 2. The electrolyte solution (200 mL) was kept at 15 °C and constantly mixed with a magnetic stirrer. The anodic oxidation was performed under an impulse current (Figure 12) up to a fixed voltage with use of a high-voltage power supply (PWR 800H, Kikusui, Japan). The method consisted of an initial galvanostatic stage (constant current density; j = 100 mA/cm^2^), which was performed until one of the three limiting anodization voltages was reached (Table 2). At this point of the procedure, the process was shifted to voltage control (potentiostatic) and the current density dropped with time. PEO treatment time was 10 min. After PEO anodized specimens were rinsed with distilled water and the samples were dried on air.

### 3.3. Surface Analysis

#### 3.3.1. SEM and EDX Analysis

The morphology analysis of the obtained coatings was conducted by scanning electron microscopy (SEM, Phenom ProX, Phenom-World BV, Eindhoven, the Netherlands, accelerating voltage = 25 kV). The pore size and pore distribution were described by the Trial Version Image-Pro 10.0.7 software. The elemental composition of the oxide coatings was characterized by an energy-dispersive X-ray spectrometer attached to the SEM.

The oxide layers’ cross-sections were determined to study the morphology and thickness of the PEO coatings. PEO modified samples were embedded in epoxy resin (Eposir F 740 + Ipox EH 2260) and then placed in a vacuum unit for 3 min before drying. Then, all samples were air-dried for 24 h and ground with a grinding machine (Einhell TH-US 400, 1440 rpm) using abrasive papers (Hermes BW114) with 400, 1000, 1200 and 1500 granulation. The morphology and chemical composition of the oxide layer structures were studied using scanning electron microscopy (SEO-SEM Inspect S50-B, FEI, Brno, Czech Republic; accelerating voltage-15 kV) paired with an energy-dispersive X-ray spectrometer (AZtecOne with X-MaxN20, Oxford Instruments plc, Abingdon, UK).

#### 3.3.2. Surface Roughness

Surface roughness was determined by the tactile stylus method. 2-dimensional profile characteristics were obtained over a length of 10 mm, using a surface roughness tester (Surftest SJ-301, Mitutoyo, Kawasaki, Kanagawa, Japan). The arithmetic mean of the sum of roughness profile values (Ra), mean peak-to-valley height (Rz) were examined in triplicate.

#### 3.3.3. Contact Angle Measurement

Contact angle (CA) provides information about the hydrophobicity or hydrophilicity properties of the surface. The wettability characteristics of the coating were made using a video-based optical contact angle-measuring instrument (OCA 15 EC, Series GM-10-473 V-5.0, Data Physics, Filderstadt, Germany). The optical analysis of ultra-pure water drops about 0.5 μL placed on a solid surface was conducted in five different positions and the average value was noted.

#### 3.3.4. Photoluminescence Analysis

Photoluminescence (PL) spectra of the PEO modified samples were measured in the range of 300–900 nm by Ocean Optics fiber spectrometer (HR4000, Largo, FL, USA). PL was excited by the Nd:YAG laser (266 nm, 20 mW, CNIlaser, Changchun, China).

### 3.4. Long–Term Immersion Test in SBF

Immersion test was conducted in a simulated body fluid (SBF) which has similar ion concentrations to human blood plasma (Table 2). All the samples were immersed separately in 50 mL of SBF at 37 °C (initial pH value of 7.4 ± 0.02). The immersed samples were placed in fresh SBF without agitation every 2 days to keep a stable chemical composition and pH value of the SBF. A wide range of immersion times including 1, 3, 7 and 63 days were selected. The degradation rates of the samples were evaluated by weight loss measurements and surface morphology analyzes. The samples were weighed before the test and after each period of immersion.

Scanning electron microscope SEO-SEM Inspect S50-B (FEI, Brno, Czech Republic) was used to describe the surface characteristics after the immersion test. The chemical composition of the surface before and during the examination was evaluated by an energy-dispersive X-ray spectrometer AZtecOne with X-MaxN20 detector (Oxford Instruments, Abingdon, Oxfordshire, UK) attached to the SEM.

### 3.5. Cell Culture

The U2OS cells were used to evaluate the biocompatibility of new coated Mg implants. Mg implants were sterilized in 70% ethanol overnight and placed on a 24-well culture plate. Implants were immersed in DMEM overnight to equalize pH. cells were seeded on each sample and in the wells without samples (as positive control) at a cell density of 10⁴ cells per well as described in [45]. Alamar blue colorimetric assay used to measure cell viability in 1, 3 and 5 days after cell seeding. The absorbance was measured using a Multiskan FC (Thermo Fisher Scientific, Waltham, MA, USA) plate reader at wavelengths of 570 and 595 nm. All experiments were repeated 3 times.

### 3.6. In Vitro Antimicrobial Activity Assay

The antibacterial activity of the samples was assessed based on the inhibition of bacterial adhesion. The provided bacteria (*S. aureus*) by the National Collection of Microorganisms (Institute of Microbiology and Virology NASU, Kyiv, Ukraine) was used as targeted bacteria. The bacterial strain was grown in a tube containing Tryptic soy broth (TSB) at 37 °C overnight. The sterile medium was inoculated with a bacterium strain to the concentration of 10^8^ by densitometer DEN-1B (Biosan SIA, Rīga, Latvia) and adjusted to a final density of 1 × 10^6^ colony forming units (CFUs)/mL. Once obtained, 2 mL of bacterium suspension was placed over the samples (PEO treated and non-coated specimens) and incubated in static conditions at 37 °C for 2, 4, 6 and 24 h. The culture broth alone and bacterium suspension in initial concentration in the absence of the samples served as controls. At each time point, the specimens were removed and gently washed three times with 2.0 mL of sterile phosphate buffer saline (PBS; pH = 7.4) to remove the unattached bacterial cells and placed into sterile plastic tubes with 1.0 mL of sterile PBS. Afterward, ultrasonication of the samples in an ultrasonic bath (B3500S-MT, Branson Ultrasonics Co., Shanghai, China) was provided for 1 min to dislodge the adherent bacteria. A total of 10 μL aliquots from each tube were transferred to agar plates with Muller–Hinton agar and spread using the streak plate technique. After overnight incubation at 37 °C counting of survivors was conducted. All the experiments were carried out under aseptic conditions in triplicate.

## 4. Conclusions

Plasma electrolytic oxidation of pure Mg in silicate-based baths provided the formation of low dissolution porous oxide coating with porosity that increased in Ca(OH)_2_-containing electrolyte (C2-type). Ca(OH)_2_ was proved to be a crucial component for oxidation and surface modification of Mg implants, treated with the PEO method. The addition of Ca^2+^ ions resulted in more intense oxidation of the Mg surface and growth of the oxide coating with a higher active surface area. Anodization of Mg in the proposed electrolytes decrease degradation mode and strongly enhanced biocompatibility. Ca(OH)_2_-containing electrolyte provided a more suitable environment for cell proliferation but demonstrated a lack of antibacterial activity. In conclusion, it should be noted that PEO of Mg alloy in Ca(OH)_2_-containing silicate-based bath provided the formation of stable biocompatible oxide coatings that could be used in the development of commercial degradable implants.

## Figures and Tables

**Figure 1 molecules-26-02094-f001:**
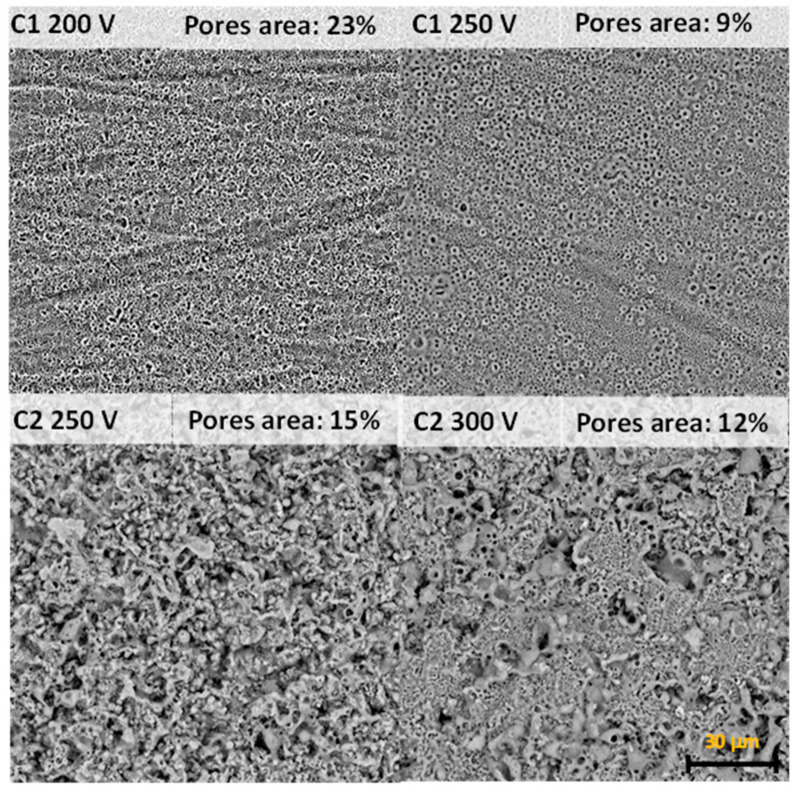
SEM planar view and pores area of the Mg samples surface after PEO process.

**Figure 2 molecules-26-02094-f002:**
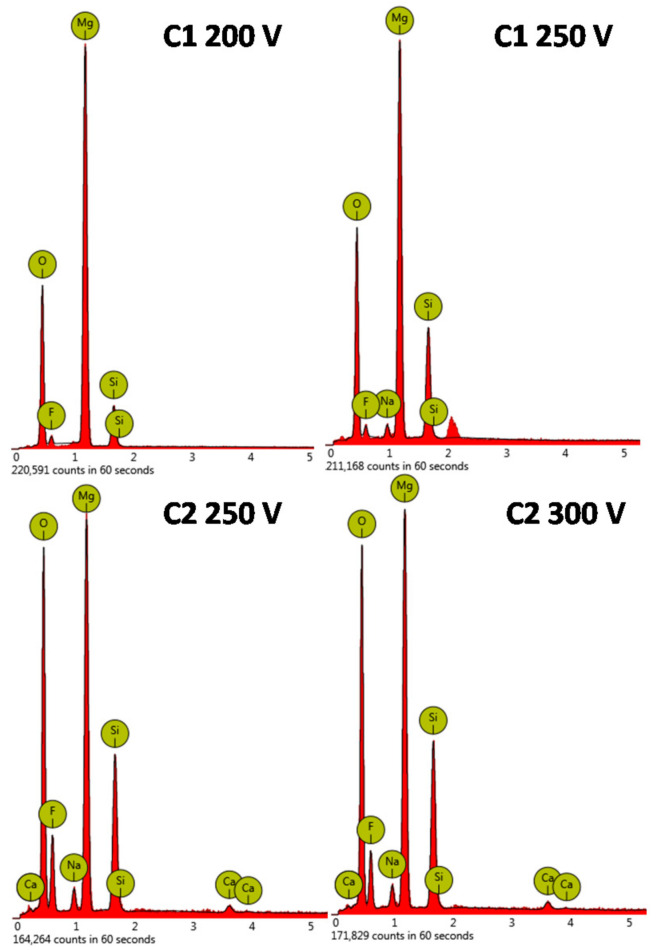
The EDX analysis of the Mg samples after PEO process.

**Figure 3 molecules-26-02094-f003:**
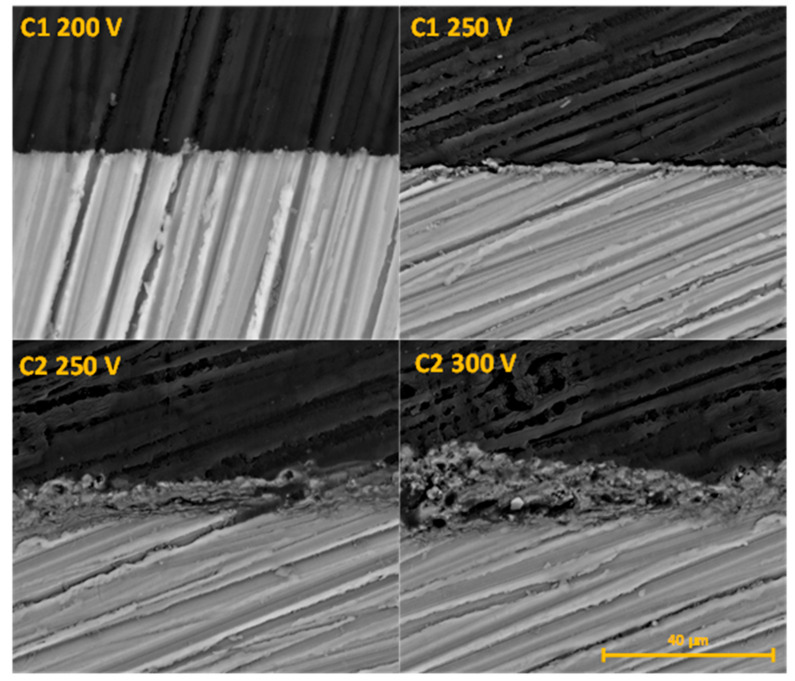
The cross-section analysis of the Mg samples after PEO process.

**Figure 4 molecules-26-02094-f004:**
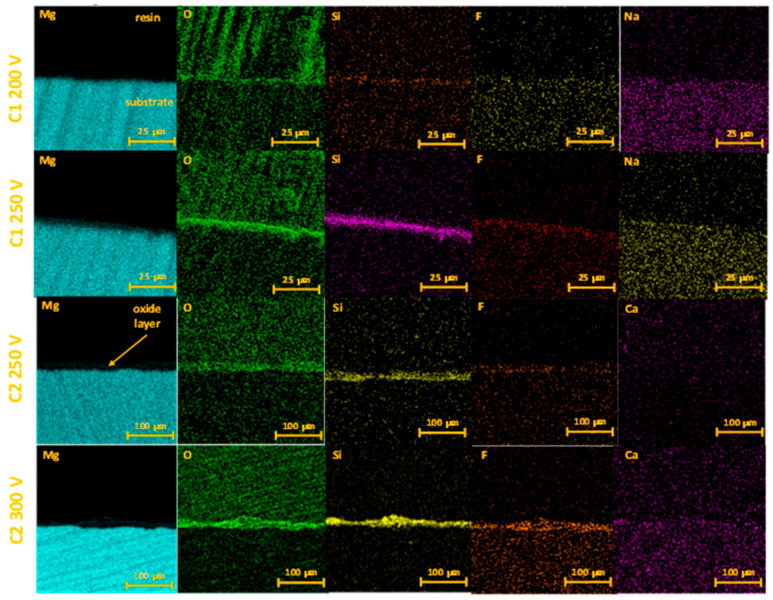
The EDX mapping of anodized Mg samples cross-section.

**Figure 5 molecules-26-02094-f005:**
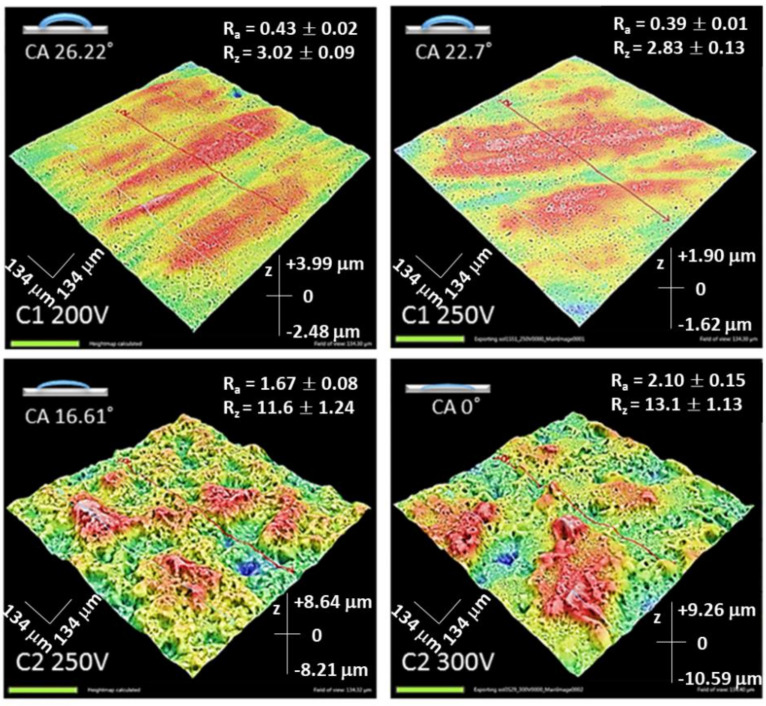
The 3D topographical maps of the anodized Mg samples as well as surface roughness and CA parameters of the coatings.

**Figure 6 molecules-26-02094-f006:**
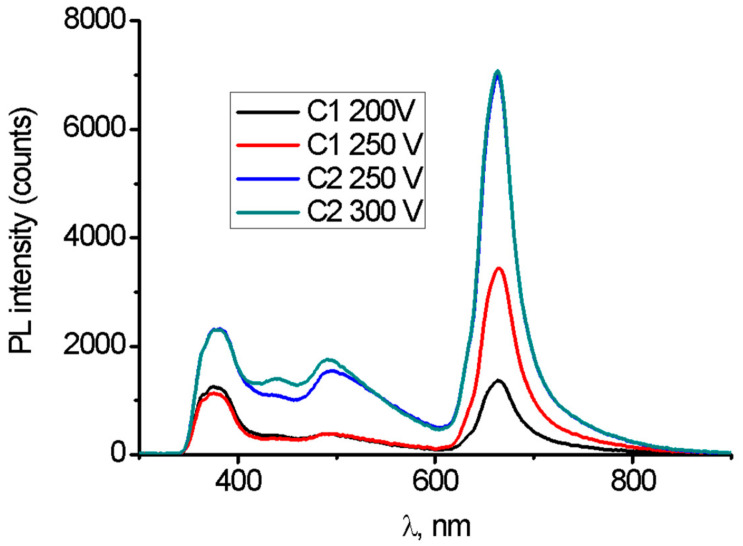
Photoluminescence of the anodized Mg samples.

**Figure 7 molecules-26-02094-f007:**
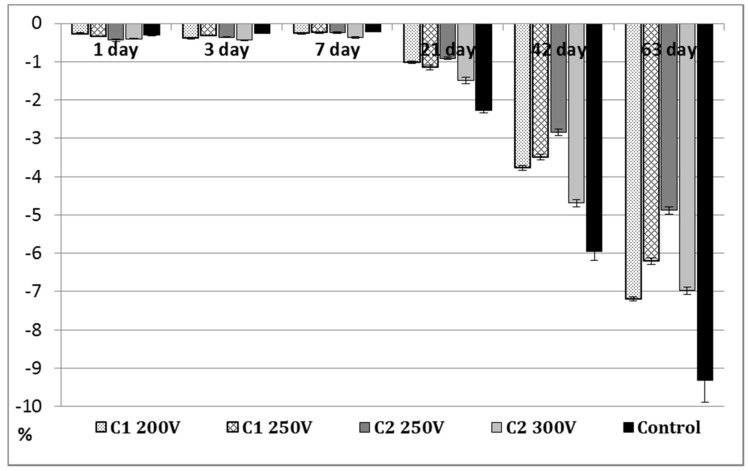
Weight loss of the anodized Mg samples.

**Figure 8 molecules-26-02094-f008:**
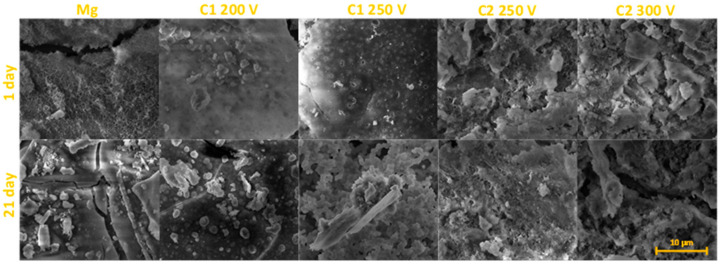
The SEM images of selected magnesium samples after immersion in the SBF solution.

**Figure 9 molecules-26-02094-f009:**
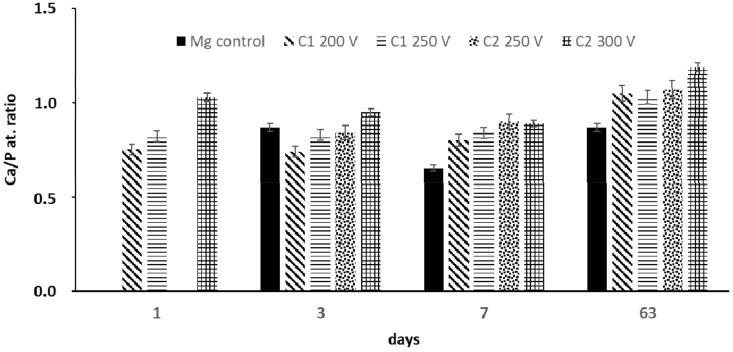
The Ca/P atomic ratio on the anodized Mg samples during the long-term immersion test.

**Figure 10 molecules-26-02094-f010:**
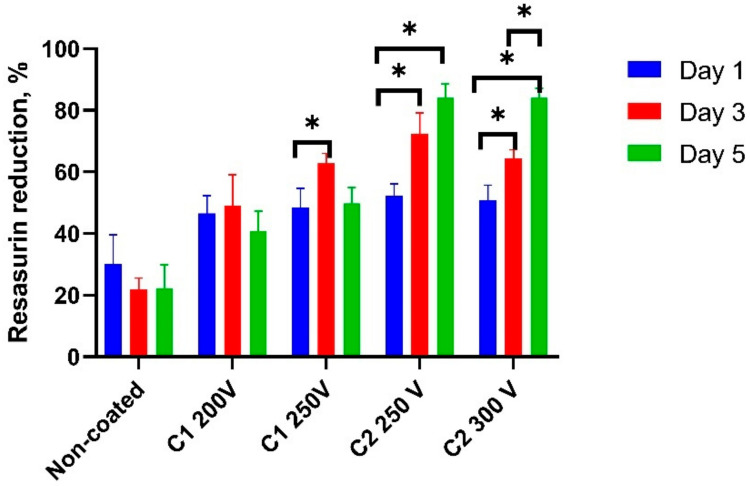
U2OS cell viability during 5 days of co-cultivation over the Mg samples (non-coated and PEO-treated). asterisks indicate a significant difference (**p* < 0.05).

**Figure 11 molecules-26-02094-f011:**
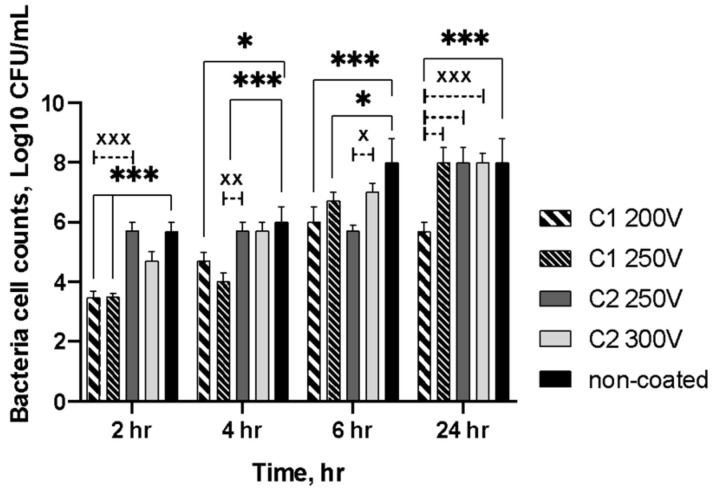
Growth and reduction of *S. aureus* on the surfaces of the samples at various time intervals; a significant difference are indicated between control and treated specimens (**p* < 0.05; ****p* < 0.0005) and between different treated samples (x *p* < 0.05; xx *p* < 0.005; xxx *p* < 0.0005).

**Figure 12 molecules-26-02094-f012:**
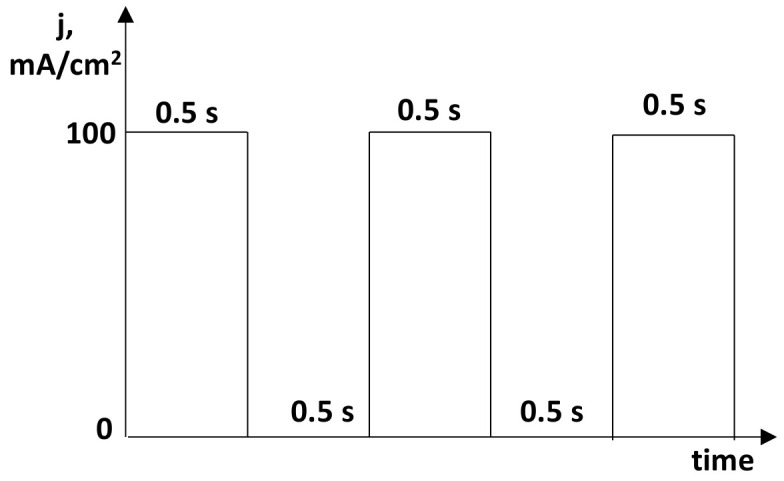
A schematic illustration of impulse current that was used.

**Table 1 molecules-26-02094-t001:** The EDX analysis of the Mg samples after the PEO process.

Sample	Anodizing Voltage, V	Detected Elements	Si/Mg at. Ratio	F/Mg at. Ratio	Ca/Mg at. Ratio
C1	200	O, Mg, F, Si	0.26 ± 0,02	0.06 ± 0.01	-
	250	O, Mg, F, Si, Na	0.52 ± 0.05	0.08 ± 0.01	-
C2	250	O, Mg, F, Si, Na, Ca	0.54 ± 0.04	0.43 ± 0.03	0.03 ± 0.01
	300	O, Mg, F, Si, Na, Ca	0.60 ± 0.05	0.31 ± 0.02	0.03 ± 0.01

**Table 2 molecules-26-02094-t002:** The chemical composition of the anodizing baths and the anodizing parameters for the PEO process.

Electrolyte Label	Concentration of Electrolyte Components; g dm^−3^				Anodizing Voltage
	Na_2_SiO_3_	NaF	NaOH	Ca(OH)_2_	
C1	10	5	10		200, 250
C2	10	5		10	250, 300

## Data Availability

The data presented in this study are available on request from the corresponding author.

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
