# Peer review of "Bioactivity Performance of Pure Mg after Plasma Electrolytic Oxidation in Silicate-Based Solutions"

_molecules, 2021, doi:10.3390/molecules26072094_

Round 1

Reviewer 1 Report

The article presents the research results of the effect of the composition of the electrolyte used for the PEO process on the properties of the formed coatings. In my opinion the article needs major revision. 1) Abstract, lines 28 and 29, the sentence is not complete. 2) Introduction, line 49, the phrase 'permanent metals' is wrong. 3) 2.1. Coatings characterization, lines 103 and 104, how the PEO voltage was changed?  In Secton Materials and Methods it wass pointed that the anodic oxidation was performed at galvanostatic mode for 10 min. 4) Section 2.2 Long-term corrosion test, there is not presented any corrosion perphormance. Weight loss parameter is not characterize  the corrosion process in this case. Magnesium corrosion results only partly in weight loss of the sample, at that the precipitation of the corrosion products and hydroxyapatite is occurred. 5) Section 2.3. Antibacterial properties, lines 285 and 286,  both C1 anc C2 coatings are hydrophilic, the contact angle is less than 90 deg. 6) Section 3.2. Plasma Electrolytic Oxidation (PEO), line 302, please clarify the impulse mode parameters, namely, frequency, duty cycle etc.

Author Response

Dear Reviewer,

Thank You very much for help in our article improvement.

1) Abstract, lines 28 and 29, the sentence is not complete.

The sentence was corrected.

2) Introduction, line 49, the phrase 'permanent metals' is wrong.

The sentence was changed.

3) 2.1. Coatings characterization, lines 103 and 104, how the PEO voltage was changed?  In Secton Materials and Methods it was pointed that the anodic oxidation was performed at galvanostatic mode for 10 min.

During the PEO process voltage grew up to a fixed value, what is typical for this type of process. Additional information was added in a methods part.

4) Section 2.2 Long-term corrosion test, there is not presented any corrosion perphormance. Weight loss parameter is not characterize  the corrosion process in this case. Magnesium corrosion results only partly in weight loss of the sample, at that the precipitation of the corrosion products and hydroxyapatite is occurred.

Thank you for this important note. We provide correction of highlights. The word “corrosion” was removed. Phrase “Long-term corrosion test” was changed to “Long-term immersion test in SBF”, and all results were described concerning to weight loss process.

5) Section 2.3. Antibacterial properties, lines 285 and 286,  both C1 anc C2 coatings are hydrophilic, the contact angle is less than 90 deg.

Answer. Dear Reviewer, definitely, both surface types are hydrophilic and we provide correction to the text according this statement.

6) Section 3.2. Plasma Electrolytic Oxidation (PEO), line 302, please clarify the impulse mode parameters, namely, frequency, duty cycle etc.

Additional information was added to the methods section.

Reviewer 2 Report

The  subject is topical, paper is clearly written and  well organized having references properly chosen and up to date. Unfortunately the novel character is not well sustained in the subchapter introduction  and the paper objective has not entirely reached due to few tests for corrosion behavior and for cell response.

In order to be published in my opinion is a need for major revision taking into account the following 

a) to present better the novelty,

b)  to performed more corrosion experiments  I do  propose   at least Tafel methods  (leading to corrosion rate) and /or electrochemical impedance spectroscopy ( EIS)

c) more tests for cells behavior as well

Author Response

Dear Reviewer,

Thank You very much for help in our article improvement.

1) to present better the novelty

Additional information was added to our manuscript, and aim of the work was clarified.

2) to performed more corrosion experiments  I do  propose   at least Tafel methods  (leading to corrosion rate) and /or electrochemical impedance spectroscopy ( EIS)

It is very interesting idea, however in present article we focused on material and biological properties of Mg after PEO process. We are going to do electrochemical investigations of magnesium and it will be an aim of our next paper.

3) more tests for cells behavior as well

Thank you for this comments. It is very important to understand cell-biomaterial interaction but the aim of current study was complex evaluation of new coating to meet general requirement for new biomaterials. Resazurin reduction test is a standard test for toxicity and proliferation assessment. Taking into account possible Mg release, and interaction of coating with fluorescent dye we did not perform fluorescent staining. To use other metabolic assays, it is requires a big set of samples that was not possible in current research. We have find “optimal coating” that will use in following research to understand or basic principles of cell-coating interaction.

Reviewer 3 Report

Dear authors,

  thank you for your submission. It was an interesting lecture. I have a list of corrections to ask you:

  • Table I: EDX is a semi-quantitative technique with a huge statistical dispersion. For each result please provide the standard deviation;
  • Figure 1: I honestly can't read the text in this figure, it's too small in A4 format. Even at 200% magnifications, the EDX labels are still unreadable;
  • Figure 4: please provide a color bar legend for these maps;
  • Figure 7: all these bars need statistical dispersions;
  • Figure 6: where is it?
  • Figure 8: again, statistics are necessary to validate this kind of results;

Other notes:

  • There seem to be a preferential orientation of the structures in Fig. 1 (C1 200V and C1 250V) but I can't understand if it's real or just pareidolia. On my screen, at 200% magnifications, I see a sort of "grid" with about 20 microns of size. It might be a image compression effect due to the low resolution;
  • The Photoluminescence results of Figure 5 are really interesting. It surprises me that there is no drastical change in the "shape" of the emission, meaning that the defects of all four samples have the same origin. Moreover, I expected the light scattering should caused by surface roughness to affect the emission. It would have been useful to compare them to a more conventional magnesium oxide, but I can't find any good reference for you;
  • "Long-term immersion test in SBF solution characterizes the corrosion response of the surface and its ability to induce the precipitation of the apatite layer" I do understand what you imply, but it's a stretch to call it "apatite layer", as it might sound like it's bone-like hydroxyapatite. Without the assistance of cells it should basically be just Calcium oxides and calcium phosphates, I guess. To call it "apatite" you need to show a diffraction pattern;
  • The authors state that the PEO-treated samples "underwent a visible corrosion process". This statement needs to be supported by SEM images;
  • "which demonstrated the toxic effect of the 233 corrosion products on cell viability" this is quite a strong claim. You are basically saying that MgO or Mg(OH)2 are toxic (to osteosarcoma cells). This seems to go against most positive results in literature and will require a detailed explanation;
  • Always always always remember to write bacteria names in italic;
  • Bacteria results don't really seem to follow a trend, or at least not one I can clearly see. Why is C1 200V better than C1 250V at 24h? Why is C2 250V basically comparable to the control at 2 hours, much lower at 6 hours and again comparable at 24 hours? I think CFU alone is not sufficient to provide a detailed discussion of the mechanisms, you should either add a different experiment or revise the claims.

Author Response

Dear Reviewer,

Thank You very much for help in our article improvement.

1) Table I: EDX is a semi-quantitative technique with a huge statistical dispersion. For each result please provide the standard deviation;

Completely agree with Reviewer. We used an EDX analysis as a preliminary tool for determination of elements in oxide coatings. We decided to show results as atomic ratios, not directly calculated chemical composition. The standard deviation was added to our results for all EDX measurements.

2) Figure 1: I honestly can't read the text in this figure, it's too small in A4 format. Even at 200% magnifications, the EDX labels are still unreadable;

Figure 1 was divided into two figures, so, the quality is better now.

3) Figure 4: please provide a color bar legend for these maps;

The legends were added.

4) Figure 7: all these bars need statistical dispersions;

The statistical dispersion was added.

5) Figure 6: where is it?

There wasn’t figure 6. Sorry. But now it is.

6) Figure 8: again, statistics are necessary to validate this kind of results;

See previous answer.

7) There seem to be a preferential orientation of the structures in Fig. 1 (C1 200V and C1 250V) but I can't understand if it's real or just pareidolia. On my screen, at 200% magnifications, I see a sort of "grid" with about 20 microns of size. It might be a image compression effect due to the low resolution;

See previous answer.

8) The Photoluminescence results of Figure 5 are really interesting. It surprises me that there is no drastical change in the "shape" of the emission, meaning that the defects of all four samples have the same origin. Moreover, I expected the light scattering should caused by surface roughness to affect the emission. It would have been useful to compare them to a more conventional magnesium oxide, but I can't find any good reference for you;

Thank you for your question. According to the papers, reported on PL in MgO we concluded that emission is explained by defects. The components, added to solution within PEO treatment might not have PL properties. Two main differences between PEO prepared samples involved the applied voltage and additives what enhanced oxidation. We suppose that more defects were formed under higher applied voltage. Surface roughness might be a reason for light scattering. However, the higher voltages induced higher surface roughness.

9) "Long-term immersion test in SBF solution characterizes the corrosion response of the surface and its ability to induce the precipitation of the apatite layer" I do understand what you imply, but it's a stretch to call it "apatite layer", as it might sound like it's bone-like hydroxyapatite. Without the assistance of cells it should basically be just Calcium oxides and calcium phosphates, I guess. To call it "apatite" you need to show a diffraction pattern;

Thank you for this comment. Your reviews are most helpful. All the changes are marked in yellow color. The phrase “apatite layer” was corrected to “calcium phosphate deposits”.

10) The authors state that the PEO-treated samples "underwent a visible corrosion process". This statement needs to be supported by SEM images;

Thanks you for your comment. A new figure described a visible corrosion process was added.

11) "which demonstrated the toxic effect of the 233 corrosion products on cell viability" this is quite a strong claim. You are basically saying that MgO or Mg(OH)2 are toxic (to osteosarcoma cells). This seems to go against most positive results in literature and will require a detailed explanation;

Thank you for this point. The direct application of Mg and its oxides are not toxic in culture media in low concentration. Due to intensive corrosion Mg and its salts and oxides can change the pH of media (shift to acidic ones) and significantly affect viability. It is very well known fact that described in [38-39]. We discussed this fact in the text.

12) Always always always remember to write bacteria names in italic;

Thank you for this comment – we provide correction through all the text. 

13) Bacteria results don't really seem to follow a trend, or at least not one I can clearly see. Why is C1 200V better than C1 250V at 24h? Why is C2 250V basically comparable to the control at 2 hours, much lower at 6 hours and again comparable at 24 hours? I think CFU alone is not sufficient to provide a detailed discussion of the mechanisms, you should either add a different experiment or revise the claims.

Thank you for this comment. C1 200V sample retained antiadhesive behavior better than all other specimens up to 24h due to the lowest hydrophilic properties as its CA was the biggest. Indeed, during the test, C2 250V was at the same level or lower than control. This sample showed the ability to restrain bacterial colonization at 2h, 4h, and 6h time points. In contrast, the control sample showed a steady increase of bacterial cell amount on its surface. We agree that CFU evaluation alone is not enough to explain the obtained results, but we discussed the mechanisms considering the effects of surface roughness and hydrophilicity.

Round 2

Reviewer 1 Report

Authors improved significantly the manuscript, but it is necessary to correct two points. 1) According to Reviewer's 3 comments  bacteria names should be write in italic, but in Abstract section bacteria name was not write in italic.  2) Again Secttion 3.2. Plasma Electrolytic Oxidation (PEO), it is inpossible to achive the different voltage at the same time treatment in single galvanostatic mode. One of parameters should be changed  the current density or treatment time.

Author Response

Dear Reviewer,

thank You very much for comments. Bacteria name in an abstract is in italic now. Additional information about PEO process was added:

"The method consisted of an initial galvanostatic stage (constant current density; j = 100 mA/cm2), which was performed until one of the three limiting anodization voltages was reached (Table 2). At this point of the procedure, the process was shifted to voltage control (potentiostatic) and the current density dropped with time. "

Reviewer 2 Report

The revised manuscript was very little improved and new experiments to enhance scientific level of manuscript has no been performed. However, I do consider that major revision is required introducing new data about electrochemical and biological behaviour of plasma electrolytic oxidation coating in silicate-based solutions on Mg.  I do believe that revised form  is not reaching the journal Molecules level.

Author Response

Thank you very much for your comments. For sure, we did not perform state of the art biological assessment of new silicate-based coating. Taking into account that the Mg alloy must corrode during the 8-12 weeks after the implantation, the main function of coating is to decrease initial metal degradation. The main biological requirement for coating is high biocompatibility and non-toxic nature of degradation products. In our research we provide complex assessment of new coating from structural, chemical, and biological sides, including cell toxicity and antibacterial properties. These investigations allow to select optimal parameters for PEO to produce stable, porous and bioactive coating. We clearly demonstrate high biocompatibility compared to non-treated metal and discussed possible mechanisms. Additionally, we firstly demonstrated ability to decrease bacteria adhesion that could be useful in orthopedic practice. We hope, that our complex assessment of new coating open perspectives for future development of new Mg implants. As we pointed in a previous answer at this moment we don’t see the need of additional investigations performance. In the case of electrochemical measurements we will not show any new knowledge. It is known that corrosion resistance of Mg after the PEO process is bigger in comparison to a pure metal. If we want to focus on corrosion properties of Mg after PEO we should plan a new set of broad investigations (including long-term corrosion and advanced analysis of corrosion products).

So, in our opinion aim of the work was achieved and additional experiments are not necessary.

Reviewer 3 Report

The authors replied to my queries and I'm ok with their answers.

Author Response

Thank You.

Round 3

Reviewer 2 Report

In my opinion the paper has not enough experimental data having few data about biological part and not enough about other  characterization such as corrosion and ions release My proposal in the first review was to introduced corrosion experimental data able to be exploited to compute porosity from polarization resistance in order to have quantified information about that. In your answer you declared that "It is very interesting idea, however in present article we focused on material and biological properties of Mg after PEO process. We are going to do electrochemical investigations of magnesium and it will be an aim of our next paper".

Unfortunately the biocompatibility  part is not complete having only few data about cells viability in a moment when aspects are  in great development with NO, ROS and more other experiments able to say something about mechanism

Moreover it is very confusing your new  reponse  about the need of corrosion experiments afirming with an indicated reference that such experiments are done in the literature and "in this situation  aim of the work was achieved and additional experiments are not necessary". But the reference is about Mg alloys which is some thing else your paper being about pure Mg.